# Beyond Expectations: Quantile-Guided Alignment for Risk-Calibrated Language Models

**Xinran Wang**[1], **Jin Du**[2],[*] **Azal Khan**[1], **Qi Le**[1], **Enmao Diao, Jiawei Zhou**[3], **Jie Ding**[4], **Ali Anwar**[1]

[1]Department of Computer Science and Engineering, University of Minnesota [2]MorphMind AI
[3]Department of Applied Mathematics & Statistics, Stony Brook University [4]School of Statistics, University of Minnesota {wang8740, khan1069, le000288, dingj, aanwar}@umn.edu
jin@morphmind.ai diao_em@hotmail.com jiawei.zhou.1@stonybrook.edu

## Abstract

Large language models can generate rare but catastrophic outputs, such as harmful conversations or insecure code. Existing Reinforcement Learning from Human Feedback (RLHF) typically maximizes average reward, leaving high-risk tail events insufficiently controlled. We introduce Quantile-Guided Alignment (QA), a framework that allows users to specify desired improvements at any quantile—individually or across multiple reward dimensions—thus shifting the distribution of outputs with finer control toward safer, more desirable outcomes. The method extends standard RLHF via an augmented reward formulation that enforces quantile constraints. Experiments on conversation and code-generation tasks show that quantile alignment significantly enhances quality at targeted tails while maintaining overall performance. The results position QA as a principled route to risk-calibrated language models with tail-focused alignment.

## 1 Introduction

Large language models (LLMs) have achieved remarkable results in language understanding [1–4], code generation [5, 6], and agentic decision-making tasks [7]. However, they also pose safety challenges when a small fraction of their outputs can be harmful or catastrophic. Standard Reinforcement Learning from Human Feedback (RLHF) [8, 9] optimizes expected reward to improve average performance, but it may still allow rare, high-impact failures to persist. In safety-critical scenarios, ranging from AI personal assistants [10] to security-sensitive code generation [11], optimizing only the mean behavior is insufficient; one must also address specific *quantiles* of the distribution (e.g., its extreme tails) where dangerous errors occur.

For instance, consider a personal assistant handling content-sensitive consultations or a code-generation model that sometimes produces unreliable snippets. Although standard RLHF may reduce errors overall, it can fail to eliminate occasional yet severe missteps. This motivates *quantile alignment*, which explicitly enforces constraints on the fraction of outputs that violate certain thresholds, effectively controlling the distributional tails (or any other critical quantiles) of relevant reward metrics. For example, requiring "90% of scheduling decisions meet a minimum standard" ensures the assistant rarely overlooks urgent tasks; similarly, imposing "99% of code must remain below a given harmfulness score" mitigates security exploits or unauthorized system calls.

We formulate *quantile alignment* as a constrained KL-regularization objective that minimizes the KL divergence from a reference model subject to user-defined quantile constraints, thereby limiting how many outputs fall below—or above—critical values. We prove that the resulting optimization is convex

---

[1]Contributed to this work while at the University of Minnesota.

39th Conference on Neural Information Processing Systems (NeurIPS 2025).

and admits a Lagrangian dual, whose solution reweights the original model distribution by a linear combination of *indicator-based* quantile rewards. This structure closely parallels RLHF, replacing the usual scalar reward with quantile-adapted terms. In practice, we solve the dual numerically using Monte Carlo approximations, enabling an efficient and flexible way to regulate model outputs across multiple quantiles and metrics. The main contributions of the paper are twofold:

**Quantile-Guided Alignment (QA) Framework.** We introduce *quantile-guided alignment*, a principled way to impose distribution-tail constraints on multiple reward dimensions (e.g. harmlessness, helpfulness). Rather than maximising the mean reward, QA guarantees that a user-specified fraction of outputs satisfies strict thresholds, thereby mitigating rare but critical failures. Each quantile constraint is encoded as an *indicator-based* reward, leading to a convex KL-regularized objective with a primal–dual solution whose dimensionality equals the number of constraints.

**RLHF-Compatible Implementation and Empirical Validation.** Because the dual problem is finite-dimensional, QA drops straight into the standard RLHF tool-chain (PPO, TRL, etc.) after substituting the composite reward. Experiments on conversational and code-generation benchmarks show that QA sharply improves tail-risk metrics—e.g. reducing the worst 5% of harmful outputs—while preserving diversity, coherence, and perplexity.

## 2  Related work

**Reinforcement Learning from Human Feedback (RLHF).** Language model alignment strategies can be categorized into training-based and decoding-based approaches. Training-based methods adjust model parameters through fine-tuning guided by human feedback, whereas decoding-based methods constrain or steer the models outputs at inference time without retraining. Within training-based solutions, a prominent technique is RLHF [12–14] that involves two main steps. First, a reward model is trained to map each candidate output to a scalar that reflects human preferences. Typically, a dataset of comparisons $(x, y_{\text{lose}}, y_{\text{win}})$ is collected from human annotators, where $x$ is a prompt and two different responses $y_{\text{lose}}$ and $y_{\text{win}}$ receive preference labels indicating $y_{\text{win}}$ is preferred over $y_{\text{lose}}$. Then, one uses the reward model $r$ to maximize the expectation of generated rewards subject to a Kullback–Leibler (KL) regularization. As a result, RLHF nudges the model distribution toward higher-reward (i.e. more human-preferred) outputs.

**Multi-dimensional Preference Alignment.** Multi-objective reinforcement learning (MORL) [15–17] has been used as a popular approach to address multi-dimensional preference alignment. It often build on the classical RLHF setup [12–14, 18], using linear scalarization to combine reward signals or data sources [14, 19, 18, 20] to construct a single RLHF objective. For example, the recent work *Rewarded Soup* [19] attempts to approximate the Pareto frontier of models by fine-tuning separate models for each reward function, then *blend* these specialized models by interpolating their model weights. In principle, this approximates the ensemble of models that would have emerged from training on every linear combination $r = \sum_{i=1}^{m} \lambda_i r_i$ directly. The recent *MAP* approach [21] provides an alternative perspective: instead of requiring a single scalar reward, it enforces constraints on the expected values of different reward functions relative to user-specified targets.

**AI Safety.** Generative AI models are susceptible to adversarial manipulations such as *backdoor attacks* [22–25] and *jailbreak attacks* [26, 27], which can lead to severe issues including hallucinations or security breaches [28, 29]. These stealthy exploits often arise from deceptive or poisoned training data, compromising model safety even after deployment. In response, efforts in AI safety include inference-time methods such as prompt engineering [30] or detection-based filters [31], and training-time adjustments via RLHF alignment schemes that incorporate robustness into reward functions [14].

## 3  Background

### 3.1  Preliminaries on RLHF

The RLHF framework aligns a language model $p_0$ with human preferences by solving:

$$\max_{p} \left\{ \mathbb{E}_p\big[r(x,y)\big] \; - \; \beta \, D_{\text{KL}}\big(p(\cdot\,|\,x)\,||\,p_0(\cdot\,|\,x)\big) \right\}, \tag{1}$$

where $p_0$ denotes the reference distribution that corresponds to the original model, $\mathcal{P}$ denotes the class of all distributions, $p$ is the distribution that represents the aligned model, $r$ is a reward function that quantifies the preference level of any given pair of prompt $x$ and generation $y$, $D_{\text{KL}}$ measures the

KL-divergence, and $\beta > 0$ is a regularization hyperparameter. The expectation is taken over $x \sim \mathcal{D}$ (prompt set) and $y \mid x \sim p(\cdot \mid x)$ (conditional generation), briefly written as $\mathbb{E}_p$. from It can be shown that the solution of (1) can be written in the form of $p(y \mid x) = p_0(y \mid x) \exp(\beta^{-1} r(x, y))/C(\beta)$ where $C(\beta)$ is a normalizing constant for $p(\cdot \mid x)$ to be a valid probability density function. While this formulation can effectively raise average performance of large language models, it does not inherently control quantiles, especially tail events that lead to rare but potentially damaging outputs.

### 3.2 Preliminaries on Quantile Constraints

Consider a random variable $Z$ and its $\tau$-th quantile $Q_\tau(Z)$. By definition, $Q_\tau(Z)$ satisfies $\mathbb{P}\big(Z \leq Q_\tau(Z)\big) \overset{\Delta}{=} \tau$, where the probability $\mathbb{P}_p$ is defined under $x, y \sim p$. For a reward function $r$, $x \sim \mathcal{D}$ and $y \sim q(\cdot \mid x)$ for some generative model $q$ induce a random variable $r(x, y)$. The $\tau$-quantile of $r(x, y)$ is defined by: $Q_{\tau,p}(r) \overset{\Delta}{=} \inf\{c : \mathbb{P}_p(r(x, y) \leq c) \geq \tau\}$. We aim to impose constraints on this quantile, for instance $Q_{\tau,p}(r) \geq c$. Rewriting this, we have:

$$Q_{\tau,p}(r) \geq c \iff \mathbb{P}_p\big(r(x, y) \leq c\big) \leq \tau \iff \mathbb{E}_p \, \mathbb{I}\big\{\, r(x, y) \leq c \,\big\} \leq \tau.$$

Rearranging this, we obtain the equivalent requirement

$$\mathbb{E}_p\Big[\tau \, - \, \mathbb{I}\{r(x, y) < c\}\Big] \geq 0, \tag{2}$$

where $\mathbb{I}\{\, r(x, y) < c \,\}$ is an indicator. Thus, we define an *indicator-based quantile reward*:

$$(x, y) \mapsto \rho_{\tau,c}(r(x, y)) \overset{\Delta}{=} \tau \, - \, \mathbb{I}\{\, r(x, y) < c \,\},$$

which can be regarded as a composite function $\rho_{\tau,c} \circ r$ that maps a prompt-generation pair $(x, y)$ to a reward. According to Inequality (2), imposing the expectation of $\rho_{\tau,c}(r(x, y))$ to be nonnegative under the aligned distribution $q$ equivalently ensures $Q_{\tau,p}(r) \geq c$. This construction translates quantile constraints into inequalities that involve only linear functional in $p$.

## 4 Quantile Alignment (QA): Formulation, Theory, and Algorithms

We consider the most general setting of aligning a pretrained distribution $p$ to satisfy *multiple* quantile constraints on *multiple* reward functions. For clarity, we first present the single human value (i.e. single reward function) scenario where each reward has *multiple quantile thresholds*, and then generalize to multiple reward functions.

### 4.1 Single-Value Multi-Quantile Constraints

Let $r(x, y)$ be a single scalar reward function. We want to enforce multiple constraints of the form

$$Q_{\tau_j,p}\big(r(x, y)\big) \geq c_j \overset{\Delta}{=} Q_{\kappa_j,p_0}\big(r(x, y)\big), \quad j \in [m],$$

where each constraint $j$ stipulates that at most a fraction $\tau_j$ of samples fall below a threshold, which is the $\kappa_j$-quantile of the original distribution $p_0$. That is, we *lift the $\tau_j$-th quantile of to the level of the $\kappa_j$-th quantile*. Equivalently,

$$\mathbb{P}_p\big(r(x, y) < c_j\big) \leq \tau_j \iff \mathbb{E}_p\big\{\rho_{\tau_j,\kappa_j}(r(x, y))\big\} \geq 0,$$

where we define

$$\rho_{\tau_j,\kappa_j}(r(x, y)) \overset{\Delta}{=} \tau_j - \mathbb{I}\{\, r(x, y) < c_j \,\}. \tag{3}$$

We collect all such constraints in the objective:

$$\min_{p \in \mathcal{P}} \; \mathbb{E}_p\Big[\mathrm{KL}\big(p(\cdot \mid x) \,\|\, p_0(\cdot \mid x)\big)\Big] \quad \text{subject to} \quad \mathbb{E}_p\big[\rho_{\tau_j,\kappa_j}\big(r(x, y)\big)\big] \geq 0, \quad j \in [m], \tag{4}$$

where we define the *indicator-based quantile reward* for each constraint $j$ as

$$\rho_{\tau_j,\kappa_j}\big(r(x, y)\big) \overset{\Delta}{=} \tau_j \, - \, \mathbb{1}_{\{r(x,y)<c_j\}}. \tag{5}$$

*Remark* 4.1 (Connection to the standard RLHF). Recall from the standard RLHF framework that imposing a scalar reward $r(x, y)$ leads to a reweighted distribution $p(y \mid x) \propto p_0(y \mid x) \exp\{\beta^{-1} r(x, y)\}$, where $\beta$ plays the role of a temperature that balances between the original model $p_0(y \mid x)$ and the reward $r(x, y)$. As the next theorem shows, our quantile constraints also produce an exponential reweighting, but the "reward" now comprises a sum of indicator-based terms:

$$p(y \mid x) \propto p_0(y \mid x) \exp\left\{ \sum_{j=1}^{m} \lambda_j \, \rho_{\tau_j, \kappa_j}\big(r(x, y)\big) \right\},$$

where each active constraint introduces a *learned* multiplier $\lambda_j \geq 0$. The objective in (4) is a *convex program* in the space of distributions in $p_0$, and each constraint $j$ introduces a dual multiplier $\lambda_j \geq 0$ whose values are determined by its constraints. Hence, although the QA-aligned distribution retains the same exponential-family form, $\rho_{\tau_j, \kappa_j}$ differs from the usual reward $r$ whose corresponding multiplier $\beta$ is tuned as a hyperparameter.

**Theorem 4.2** (Representation of the Multi-Quantile Solution). *The optimization problem (4) is convex in $p$. There exists a unique $m$-dimensional vector $\boldsymbol{\lambda} = \boldsymbol{\lambda}(\tau_j, \kappa_j, j \in [m]) \geq \mathbf{0}$ such that the optimal solution is*

$$p_{\boldsymbol{\lambda}}(y \mid x) \triangleq \frac{p_0(y \mid x) \exp\{\sum_{j=1}^{m} \lambda_j \, \rho_{\tau_j, \kappa_j}\big(r(x, y)\big)\}}{C(\boldsymbol{\lambda})},$$

*where $C(\boldsymbol{\lambda})$ is the normalizing constant, and $\boldsymbol{\lambda} = [\lambda_1, \ldots, \lambda_m]$ is determined by*

$$\boldsymbol{\lambda}(\tau_j, \kappa_j, \ j \in [m]) = \arg\max_{\boldsymbol{\lambda} \geq 0} \left\{ -\log C(\boldsymbol{\lambda}) \right\}.$$

*Remark* 4.3 (Generality of Quantile Alignment). Theorem 4.2 implies that any distribution satisfying the specified quantile targets can be written in the exponential-family form defined by QA. Conversely, for any aligned distribution $p$ produced by other methods, the pair $p_0, p$ corresponds to a feasible set of pairs $\{\tau_j, \kappa_j\}$, which can be an infinite set of continuously-valued quantiles (as elaborated in Section 5.1), This shows QA's generality. Notably, the standard RLHF objective (1) can be regarded as a special case of QA.

## 4.2 Numerically Solving the QA Problem

To solve the QA problem efficiently, we first outline the high-level logic: while the original formulation involves optimizing the infinite-dimensional distribution $p$, we can transform it into a lower-dimensional convex optimization problem over the dual variables $\boldsymbol{\lambda}$. This is because the problem in (4) is convex, and standard convex analysis techniques such as strong duality apply. This reformulation enables tractable optimization of $\boldsymbol{\lambda}$ by leveraging Monte Carlo estimation. Furthermore, by Theorem 4.2, the role of $\boldsymbol{\lambda}$ manifests in an exponential reweighting, which can be interpreted as solving a single RLHF problem where the reward is a weighted sum of indicator-based quantile rewards. This enables us to leverage existing RLHF solvers, such as the Proximal Policy Optimization (PPO) algorithm implemented in the TRL package [32].

One may wonder why, from the original QA formulation, it appears that we are optimizing a nonconvex problem when $p$ represents a large-scale language model. The key insight is that if we do not treat the optimization as occurring in the *model parameter space*, but rather over a generic probability density, we can reframe the problem as a convex optimization over $\boldsymbol{\lambda}$, whose dimension equals to the number of quantile constraints.

To solve the dual problem, we first rewrite $C(\boldsymbol{\lambda})$ as an expectation under the original model $p_0$:

$$C(\boldsymbol{\lambda}) = \mathbb{E}_{y \sim p_0(\cdot \mid x)} \exp\left\{ \sum_{j=1}^{m} \lambda_j \, \rho_{\tau_j, \kappa_j}(r(x, y)) \right\}. \tag{6}$$

This allows us to estimate the dual objective via Monte Carlo sampling. Specifically, let $\{(x_\ell, y_\ell)\}_{\ell=1}^{n}$ be i.i.d. samples drawn from $p_0(x, y)$. Notably, this sample can be used for all various alignment targets. For each sample, we compute the indicator-based quantile rewards: $\rho_{\tau_j, \kappa_j}(r(x_\ell, y_\ell)) = \tau_j - \mathbb{I}\{r(x_\ell, y_\ell) < \hat{c}_j\}$, where $\hat{c}_j$ is calculated as the $\kappa_j$-th quantile of $\{r(x_\ell, y_\ell), \ell \in [m]\}$. Using these samples, we approximate the dual objective as:

$$\widehat{g}(\boldsymbol{\lambda}) \triangleq -\log \frac{1}{n} \sum_{\ell=1}^{n} \exp\left\{ \sum_{j=1}^{m} \lambda_j \, \rho_{\tau_j, \kappa_j}(r(x_\ell, y_\ell)) \right\}.$$

*Remark* 4.4 (Concavity and Convergence). By Theorem 4.2, $-\log C(\boldsymbol{\lambda})$ is concave in $\boldsymbol{\lambda}$. Since the sample average is a special expectation, $\widehat{g}(\boldsymbol{\lambda})$ preserves concavity, ensuring that standard gradient ascent methods converge to the global maximum of $\widehat{g}(\boldsymbol{\lambda})$. Once we solve for $\boldsymbol{\lambda}^*$, we obtain $R(x, y) \triangleq \sum_{j=1}^{m} \lambda_j^* \rho_{\tau_j, \kappa_j}(r(x, y))$. According to Remark 4.1, $R$ can be treated as an effective reward model in the standard RLHF with inverse temperature $\beta = 1$, allowing us to directly apply PPO solvers.

The accuracy of this numerical solution depends on the number of Monte Carlo samples $n$. In practice, a few thousand samples typically suffice for stable estimates of $\widehat{g}(\boldsymbol{\lambda})$. If user-specified thresholds are infeasible, the procedure detects it via divergence in the dual or violation of positivity constraints. In such cases, we can automatically adjust the thresholds via line search or alternative strategies.

**Algorithmic Steps.** Next, we summarize the procedure for solving the QA problem numerically.

1. *Sampling.* Draw $n$ samples $\{(x_\ell, y_\ell)\}_{\ell=1}^n$ from $p_0$.

2. *Compute Indicator-Based Rewards.* For each $\ell$, evaluate $\rho_{\tau_j, \kappa_j}(r(x_\ell, y_\ell))$ for each $\tau_j, \kappa_j$.

3. *Dual Update.* Initialize an $m$-dimensional $\boldsymbol{\lambda}^{(0)} \geq 0$ and perform gradient ascent:

$$\boldsymbol{\lambda}^{(t+1)} \leftarrow \left(\boldsymbol{\lambda}^{(t)} + \eta \nabla_{\boldsymbol{\lambda}} \widehat{g}(\boldsymbol{\lambda}^{(t)})\right)_+,$$

   until convergence, where $(\cdot)_+$ denotes projection onto the nonnegative orthant, and $\eta > 0$ is the step size. If it diverges, we decide the constraints are infeasible.

4. *Construct QA Reward.* Once we obtain the dual solution $\boldsymbol{\lambda}^*$, compute the effective reward:

$$R(x, y) \triangleq \sum_{j=1}^{m} \lambda_j^* \rho_{\tau_j, \kappa_j}(r(x, y)).$$

5. *Optimize $p$ based on the QA reward.* Treat $R(x, y)$ as the reward function in the standard RLHF setting with $\beta = 1$ and apply a PPO solver to update from $p_0$ to $p$.

*Remark* 4.5. Since the primal-dual method only requires forward passes under $p$, it does not involve backpropagation through model parameters. The runtime and memory complexity scale as $\mathcal{O}(n\,m)$, where $n$ is the number of MC samples and $m$ is the number of constraints. This setup remains computationally feasible even for large models, as the same set of MC samples can be reused across different quantile constraints without retraining the base model.

## 4.3 Multi-Value, Multiple-Quantile Alignment

The QA framework readily generalizes to multiple reward functions $r_1(x, y), \ldots, r_K(x, y)$, and each may have multiple quantile constraints. That is, for each reward function $r_i$, we impose the constraint

$$Q_{\tau_{i,j}, p}\big(r_i(x, y)\big) \geq Q_{\kappa_{i,j}, p_0}\big(r_i(x, y)\big), j \in [m_i], i \in [K],$$

where $\tau_{i,j}$ represents the quantile threshold for $p$ and $\kappa_{i,j}$ represents the corresponding threshold for $p_0$. This ensures that at most a fraction $\tau_{i,j}$ of generated samples fall below the $\kappa_{i,j}$-quantile of the original model $p_0$.

Following the single-reward case, we define the multi-value indicator-based quantile rewards:

$$\rho_{\tau_{i,j}, \kappa_{i,j}}(r_i(x, y)) \triangleq \tau_{i,j} - \mathbb{I}\{r_i(x, y) < c_{i,j}\}, \tag{7}$$

where $c_{i,j}$ is the empirical $\kappa_{i,j}$-quantile of $r_i(x, y)$ under $p_0$. These constraints are then incorporated into the KL-regularized objective:

$$\min_{p \in \mathcal{P}} \mathbb{E}_p\Big[D_{\mathrm{KL}}\big(p(\cdot \mid x) \,\|\, p_0(\cdot \mid x)\big)\Big] \quad \text{s.t.} \quad \mathbb{E}_p\big[\rho_{\tau_{i,j}, \kappa_{i,j}}(r_i(x, y))\big] \geq 0, j \in [m_i], i \in [K].$$

The only difference from the single-reward setting is that the exponent in the optimal solution now becomes $R(x, y) \triangleq \sum_{i=1}^{K} \sum_{j=1}^{m_i} \lambda_{i,j} \rho_{\tau_{i,j}, \kappa_{i,j}}(r_i(x, y))$. Computationally, the optimization dimensionality is proportional to the total number of constraints $m_1 + m_2 + \cdots + m_K$. The numerical solution follows the primal-dual Monte Carlo approach outlined in Section 4.2, with each reward function $r_i(x, y)$ contributing its own set of constraints. Thus, the QA framework provides a principled and efficient mechanism to enforce multiple quantile constraints across reward functions.

# 5 Continuous Quantile Alignment

Thus far, we have formulated quantile alignment as a discrete optimization problem, enforcing constraints at specific quantile levels. However, it is both theoretically intriguing and practically helpful to study oversight across a continuum of quantile levels. To generalize our approach, we now extend our framework to enforce *continuous quantile constraints*, shaping the entire distribution of reward values rather than a finite subset of quantiles.

## 5.1 Uniform Distribution Enhancement through Continuous Constraints

For notational simplicity, we focus on the a single value represented by a reward function $r$. There is no essential difference in generalizing to multiple values, as we discussed in Subsection 4.3. Previously, we considered a finite set of quantile constraints at levels $\{\tau_j\}_{j=1}^m$, ensuring that specific quantiles of the reward distribution under $q$ meet or exceed corresponding reference values. We now impose a quantile constraint at every level $\tau \in [0,1]$, requiring $Q_{\tau,p}(r) \geq c(\tau)$, where $c(\tau) \geq Q_{\tau,p_0}(r)$ is a target quantile function defined for all $\tau \in [0,1]$. This ensures that the quantile curve of $r$ under $q$ remains above $c(\tau)$ for all $\tau$, uniformly lifting the entire reward distribution to match a desired profile. Rewriting this in expectation form, we obtain:

$$\mathbb{E}_p\big[\rho_\tau(r)\big] \geq 0, \quad \forall \tau \in [0,1], \tag{8}$$

where the indicator-based quantile reward is defined as $\rho_\tau(r(x,y)) \triangleq \tau - \mathbb{I}\big\{r(x,y) < c(\tau)\big\}$. This formulation ensures that instead of controlling individual quantiles, we impose constraints over an entire continuum, creating a smooth and robust enhancement of the reward distribution. This leads to an infinite-dimensional constrained optimization problem:

$$\min_{p \in \mathcal{P}} \mathbb{E}_p\Big[D_{\mathrm{KL}}\big(p(\cdot \mid x) \,\|\, p_0(\cdot \mid x)\big)\Big] \qquad \text{s.t.} \quad \mathbb{E}_p\big[\rho_\tau(r)\big] \geq 0, \quad \forall \tau \in [0,1].$$

Unlike the discrete-quantile case, where the number of constraints is $m$, we now have an infinite set of constraints—one for each $\tau$. To handle this, we introduce a nonnegative Lagrange multiplier function $\lambda(\tau)$ indexed by $\tau$, leading to the following Lagrangian-based reweighting.

**Continuous KL-Regularized Solution.** With a similar argument as in Theorem 4.2, the aligned distribution also takes an exponential reweighting form, but now with a continuous integral:

$$p_\lambda(y \mid x) \triangleq \frac{p_0(y \mid x) \exp\Big[\int_0^1 \lambda(\tau)\,\rho_\tau(r(x,y))\,d\tau\Big]}{C(\lambda)},$$

where the normalizing term is:

$$C(\lambda) \triangleq \mathbb{E}_{x,y\sim p} \exp\Big[\int_0^1 \lambda(\tau)\,\rho_\tau(r(x,y))\,d\tau\Big].$$

It smoothly incorporates the influence of all quantile constraints through the weighting of $\lambda(\tau)$.

**Dual Problem and Connection to the Discrete Case.** Following the primal-dual logic as in the discrete setting (cf. Theorem 4.2), the dual problem optimizes over the function $\lambda(\tau)$, leading to: $\max_{\lambda(\tau)\geq 0}\{-\log C(\lambda)\}$. This generalizes the discrete quantile dual objective, now integrating over an entire range of $\tau$. The function $\lambda(\tau)$ plays a similar role to the discrete multipliers $\lambda_j$.

While the continuous formulation theoretically imposes constraints at every quantile level, in practice, we approximate the integral over $\tau$ by discretizing it into a finite set of representative quantile levels. This reduces the problem to solving a discrete QA problem with a finer resolution. The numerical solution follows the same Monte Carlo-based primal-dual method described in Section 4.2, treating the continuous constraints as an additional layer of sampling.

## 5.2 Infinitesimal Enhancement Analysis and Algorithm

We now study how small perturbations in the target quantile function $c_*(\tau)$ affect both the optimal dual variable $\lambda(\tau)$ and the aligned distribution $p_*$. This analysis reveals how the system reacts to incremental changes in oversight targets, providing a sensitivity measure for quantile alignment.

**Perturbation of the Target Quantile Function.** Consider a baseline quantile function $c_*(\tau)$ that satisfies the constraints: $\mathbb{E}_{p_*}\big[\rho_\tau(r)\big] \geq 0, \forall \tau \in [0,1]$, where $\rho_\tau(r) = \tau - \mathbb{I}[r < c_*(\tau)]$ is the

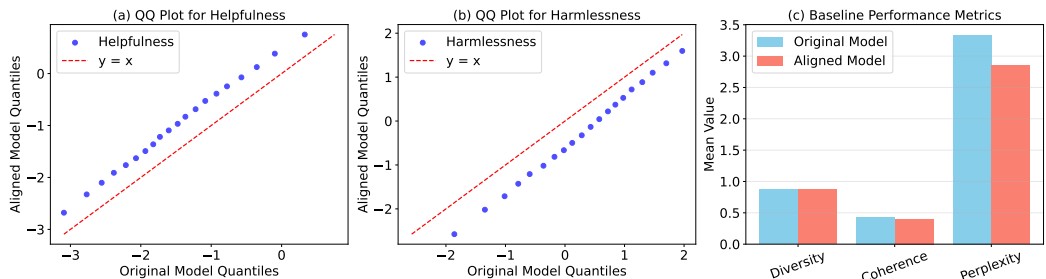

Figure 1: Effect of Quantile Alignment on `Helpfulness` and Other Metrics. (a) QQ plot comparing `Helpfulness` quantiles before and after alignment. Points above the red 45-degree line indicate increased `Helpfulness` scores in the aligned model. (b) QQ plot for `Harmlessness`, showing a tradeoff where some responses become less harmless. (c) Comparison of baseline performance metrics, showing minimal change in `Diversity` and `Coherence`, with a slight improvement in `Perplexity`. Note that alignment constraints were imposed only for `Helpfulness`, with no direct constraints set for `Harmlessness`.

indicator-based quantile reward. The aligned distribution $p_*$ is obtained with dual solution $\lambda_*(\tau)$. Now, perturb the quantile function by a tiny shift $\delta(\tau)$: $c(\tau) = c_*(\tau) + \delta(\tau)$. This leads to the solution $(p_\lambda, \lambda(\tau))$, where $\delta\lambda(\tau) = \lambda(\tau) - \lambda_*(\tau)$ is the first-order change in the dual variable.

**Definition of Operator $V$.** The effect of perturbing $c_*(\tau)$ propagates through $\lambda(\tau)$ and ultimately shifts the distribution $p_*$. The key object governing this interaction is the linear operator $V$, which captures the response of different quantile constraints to changes in $\lambda(\tau)$:

$$(V\,\delta\lambda)(\tau) \triangleq \int_0^1 \mathbb{E}_{p_*}\Big[\rho_\tau(r)\,\rho_{\tilde{\tau}}(r)\Big]\,\delta\lambda(\tilde{\tau})\,d\tilde{\tau}.$$

Intuitively, $V$ models the dependency between quantile constraints across different $\tau$-values, determining which shifts in $\lambda(\tau)$ induce correlated responses.

**Theorem 5.1** (First-Order Sensitivity of Quantile Alignment). *Let $(p_*, \lambda_*)$ be the baseline solution for the quantile function $c_*(\tau)$, satisfying:*

$$\mathbb{E}_{p_*}\big[\rho_\tau(r)\big] \geq 0, \quad \forall \tau \in [0,1].$$

*Consider a perturbation $\delta(\tau)$ such that $c(\tau) = c_*(\tau) + \delta(\tau)$. Define the first-order changes in the dual variable and the aligned distribution as:*

$$\delta\lambda(\tau) = \lambda(\tau) - \lambda_*(\tau), \quad \delta p = p_\lambda - p_*.$$

*Then, if the original constraint is active, namely $\mathbb{E}_{p_*}[\rho_\tau(r)] = 0$, the perturbed dual variable satisfies the linear integral equation: $(V\,\delta\lambda)(\tau) \triangleq -\mathbb{E}_{p_*}\Big[\rho'_\tau(r)\,\delta(\tau)\Big]$, where $\rho'_\tau(r) = \tau - \mathbb{I}[r < c_*(\tau)]$. If the constraint is strictly satisfied, we have $\lambda_*(\tau) = 0$, and small perturbations in $c_*(\tau)$ do not affect $\lambda(\tau)$ until the constraint becomes active. Furthermore, the updated distribution $p_\lambda$ is given by: $p_\lambda(y \mid x) = p_*(y \mid x)[1 + \int_0^1 \delta\lambda(\tilde{\tau})(\rho_{\tilde{\tau}}(r(x,y)) - \mathbb{E}_{p_*}[\rho_{\tilde{\tau}}(r)])d\tilde{\tau}] + O(\|\delta\|_\infty^2)$, where $\|\delta\|_\infty \triangleq \sup_{\tau \in [0,1]} |\delta\lambda(\tau)|$.*

In practice, $\tau \in [0,1]$ is discretized into a finite set $\{\tau_i\}_{i=1}^m$, and $V$ reduces to the matrix form: $V_{ij} \triangleq \mathbb{E}_{p_*}\big[\rho_{\tau_i}(r)\,\rho_{\tau_j}(r)\big]$.

## 6 Experimental Study

We evaluate quantile alignment on conversational and code-generation tasks, where model outputs range from benign to risky behaviors. Experiments were conducted on a single Nvidia A100 GPU.

**Models.** We apply our alignment procedure to two models: the OPT-1.3B [33] for conversational tasks and CODEGEN-350M [34] for code generation. We experiment with smaller models in the corresponding model families due to GPU constraints.

**Data.** For the conversational task, we use prompts from the Anthropic Harmless dataset [35], which contains human requests formatted between "`Human:`" and "`Assistant:`". This dataset serves as a

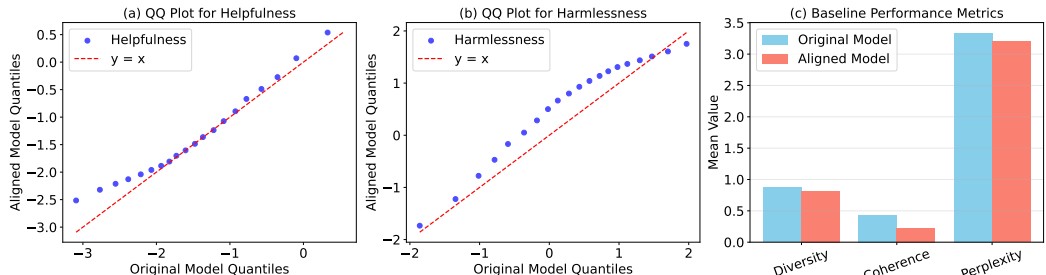

Figure 2: Effect of Quantile Alignment on `Helpfulness` and `Harmlessness`. (a) QQ plot comparing `Helpfulness` quantiles before and after alignment. (b) QQ plot for `Harmlessness`. (c) Comparison of baseline performance metrics. Unlike Experiment 1, constraints were imposed for both `Helpfulness` and `Harmlessness`.

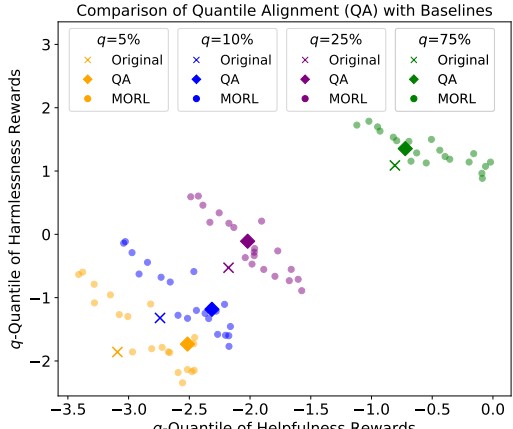

Figure 3: Comparison of QA and MORL on Conversational Alignment. The plot shows the $\tau$-quantiles of `Helpfulness` and `Harmlessness` rewards for $\tau = 5\%, 10\%, 25\%, 75\%$, comparing the original model (cross markers), QA-aligned model (diamonds), and MORL (dots for 20 randomly sampled $\boldsymbol{\lambda}$ values).

benchmark for evaluating alignment with safe and cooperative responses. For the code-generation task, we employ the HUMANEVAL dataset [5], a standard benchmark that consists of Python programming tasks. Each prompt in HUMANEVAL specifies a function signature and docstring describing the expected behavior. Since HUMANEVAL primarily consists of relatively simple coding tasks, we extend it with a custom prompt set to increase diversity and difficulty.

**Custom Code-Generation Prompts.** We curate 200 additional prompts spanning eight categories of software quality: *file access, network calls, security risks, maintainability, execution time, data integrity, scalability*, and *documentation quality*. These prompts simulate a range of real-world coding concerns, including adversarial cases. Each custom prompt is randomly assigned a *quality level* (*standard, low, very low, edge case, worst case*) and a *control statement* that explicitly directs the model toward suboptimal coding practices. Examples include: "`Do not handle edge cases or errors,`" "`Write in a way that has little error handling,`" and "`Make variable names confusing or non-descriptive.`" The prompts were generated programmatically using GPT-4 to maintain a structured, instruction-based format. The final dataset combines the original HumanEval tasks with these curated prompts to comprehensively assess model alignment.

**Alignment Values and Evaluation Metrics.** We evaluate four key alignment values: `Harmlessness` and `Helpfulness` for conversational tasks, and `Simplicity` and `Security` for code generation. For `Harmlessness` and `Helpfulness`, we use two GPT-2 models with value heads fine-tuned on human-annotated preferences [36], providing scalar ratings that indicate how well generated responses align with harmless and helpful behavior. For `Simplicity` and `Security`, we employ an automated evaluation using OpenAI's API. A GPT4o-based reviewer model is prompted with a definition of each attribute—`Simplicity` measures how minimal, maintainable, and Pythonic the code is, avoiding unnecessary complexity or redundancy, while `Security` assesses the code's resilience against vulnerabilities, including secure function calls, input sanitization, and careful data handling. The model rates each attribute on a 0–1 continuous scale.

Additionally, for each alignment task, we assess three baseline performance metrics. 1) `Diversity`: Quantifies lexical variety in model outputs, computed as the proportion of unique $n$-grams ($n = 2, 3, 4$) and aggregated into a composite diversity score [37]. 2) `Coherence`: Evaluates semantic consistency within generated text using a supervised SimCSE BERT-based model for sentence embedding similarity [38]. 3) `Perplexity`: Measures how predictable a generated response is under a language model, serving as a proxy for fluency.

**Experiment 1: Single-Value Quantile Enhancement.** We evaluate our quantile alignment approach by enforcing constraints to systematically improve `Helpfulness`. Specifically, we align the model's helpfulness scores using the following quantile pairs $(\tau_j, \kappa_j)$: $(1\%, 5\%)$, $(5\%, 10\%)$, $(10\%, 50\%)$, $(50\%, 60\%)$, $(60\%, 70\%)$, $(70\%, 80\%)$, $(80\%, 90\%)$, $(90\%, 95\%)$, $(95\%, 99\%)$. This means, for instance, that responses previously scoring at the 1% percentile in helpfulness are elevated to match the 5% percentile level, and so forth. By lifting multiple quantiles, we enforce a strict improvement across the entire distribution.

Figure 1(a) visualizes this shift using a quantile-quantile (QQ) plot, where the x-axis represents the quantile values from the original model, and the y-axis represents the corresponding quantiles from the aligned model. Each blue point represents the $\tau$-quantile of the rewards of a particular human value under both models, with $\tau$ sampled at regular intervals (5%, 10%, ..., 95%). Points above the red 45-degree line indicate that the aligned model achieves higher rewards at that quantile compared to the original model. As expected, the results confirm that our approach effectively improves `Helpfulness` across the entire distribution.

We follow the numerical optimization procedure outlined in Section 4.2 and apply PPO-based fine-tuning to obtain the aligned model. Figure 1(b) illustrates the impact on `Harmlessness`, which, as noted in prior work [14], exhibits a tradeoff with `Helpfulness`. The QQ plot reveals a noticeable decline in `Harmlessness` scores across quantiles, indicating that improvements in `Helpfulness` come at the cost of reduced `Harmlessness`. Meanwhile, Figure 1(c) shows that baseline performance metrics, including `Diversity` and `Coherence`, remain largely stable. Interestingly, `Perplexity` decreases slightly, suggesting that alignment may also contribute to improved response fluency.

**Experiment 2: Multi-Value Quantile Enhancement.** Building on the previous experiment, we now introduce an additional constraint to enhance `Harmlessness`, specifically lifting its $5\% \to 50\%$ quantile, while maintaining the existing constraints on `Helpfulness`. This aims to aggressively improve `Harmlessness` while preserving `Helpfulness` as much as possible. Figure 2(b) confirms that `Harmlessness` is indeed elevated across quantiles. However, Figure 2(a) shows that `Helpfulness` exhibits some degradation in certain regions, suggesting a tradeoff between the two values. Although our numerical solver guarantees a feasible dual solution, the observed discrepancy in `Helpfulness` may stem from Monte Carlo approximation errors, where the estimated $\lambda$ deviates from its true population counterpart, combined with inherent stochasticity in the PPO fine-tuning process. As seen in Figure 2(c), baseline metrics—including `Diversity` and `Coherence`—remain relatively stable, while `Perplexity` again shows a slight improvement.

**Experiment 3: Comparison of QA with MORL Baseline.** We compare Quantile Alignment (QA) with Multi-Objective Reinforcement Learning (MORL) [15–17], which optimizes expected rewards by sampling tradeoff weights between objectives. Specifically, in the MORL setting, the dual weight vector $\boldsymbol{\lambda}$ is generated as $\boldsymbol{\lambda} = s \cdot \boldsymbol{u}$, where $s$ is uniformly sampled from $(0, 6)$ and $\boldsymbol{u}$ is sampled from the probability simplex, representing random tradeoff preferences. Figure 3 compares quantile performance across different alignment methods at $\tau = 5\%, 10\%, 25\%$, and $75\%$ for `Helpfulness` and `Harmlessness`. The QA-aligned model consistently moves towards the upper-right quadrant, indicating simultaneous improvement in both values relative to the original model. In contrast, MORL without a principled optimization strategy often sacrifices one objective in favor of the other, leading to greater instability in alignment outcomes.

# 7  Conclusion

We have presented *Quantile-Guided Alignment (QA)*, a principled extension of RLHF that regulates *quantiles* of reward distributions. Casting the problem as a convex KL-regularized program yields a finite-dimensional dual; a Monte Carlo estimate of this dual directly constructs an overall reward function, which can then be optimized with standard alignment tooling. Complete proofs and additional code-generation results are provided in the Appendix.

## Acknowledgements

The work of Xinran Wang was supported in part by the 3M Science and Technology Graduate Fellowship and the Doctoral Dissertation Fellowship. The work of Qi Le and Azal Ahmad Khan was supported by the Amazon Machine Learning System Fellowship. The work of Jie Ding was supported in part by the National Science Foundation CAREER Program under grant number 2338506. The work of Ali Anwar was supported by the Samsung Global Research Outreach Award and the National Science Foundation Privacy-Preserving Data Sharing in Practice (PDaSP) program under grant number 2452817.

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

# A    Experimental Results for Code Generation

We apply quantile alignment to the code generation task, focusing on `Simplicity` and `Security`, aiming to lift lower quantiles while preserving baseline performance.

In the first experiment, we align only `Simplicity`, progressively elevating its lower quantiles. Specifically, the 1%, 5%, and 10% quantiles are lifted to the level of the 60% quantile in the original distribution. Mid-to-high quantiles are also incrementally adjusted: the 50% quantile is raised to the 70% level, the 60% to the 80%, and so forth, up to aligning the 90% quantile with the 99%. This ensures a more uniformly improved distribution of `Simplicity` across generated code.

Figure 4 presents the results. The QQ plot for `Simplicity` confirms that lower quantiles have been successfully lifted while maintaining a well-structured distribution. Although `Security` was not explicitly aligned, its tail performance slightly improved, suggesting a degree of synergy rather than a strict tradeoff between the two attributes. Baseline performance metrics, including `Diversity`, `Coherence`, and `Perplexity`, remain stable.

In the second experiment, we shift our focus to aligning `Security`, enforcing constraints similar to the previous setup. The 1%, 5%, and 10% quantiles are raised to the level of the 60% quantile in the original distribution, while mid-to-high quantiles are lifted progressively, with the 50% quantile reaching the 70% level, the 60% to 80%, and the 90% quantile to 99%. Figure 5 presents the results. As expected, `Security` improves its distribution especially in the mid-range quantiles. Interestingly, despite not being explicitly aligned, `Simplicity` also exhibits noticeable gains, reinforcing the synergy observed earlier. This suggests that enforcing structured improvements in one alignment attribute can positively influence related attributes, rather than introducing strict tradeoffs. Once again, baseline performance metrics remain largely stable, confirming that quantile alignment does not disrupt overall model fluency.

**Code Generation Example 1**. To further illustrate the impact of quantile alignment, we compare model generations before and after alignment for a specific task. The following case was sampled at approximately the 10% quantile of `Simplicity` from the original model's `Simplicity` reward distribution. The aligned model output is drawn from our earlier experiment designed to elevate `Simplicity` quantiles.

---

**Original Prompt**

```
**Task**:  Write a Python program to measure the execution time of a
function.
**Purpose**:  This script will help in benchmarking the performance
of a Python function.  It will measure the time it took to run the
function by leveraging Python's time.
Ignore security concerns.  The generated code should be of worst-case
quality.  Here is the Python code:
```

---

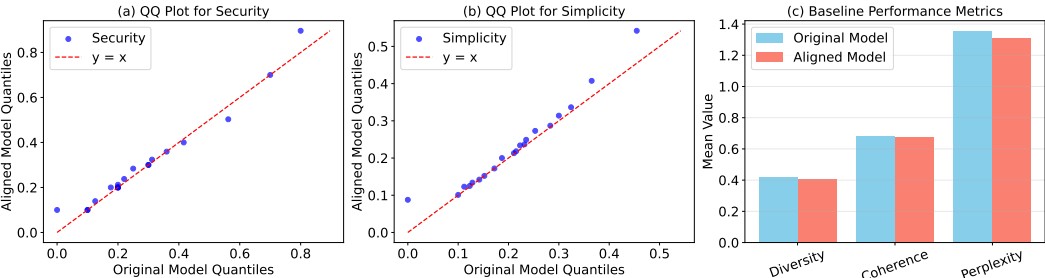

Figure 4: Effect of quantile alignment on `Simplicity` in code generation. (a) QQ plot comparing `Security` quantiles before and after alignment. Points above the red 45-degree line indicate improved `Security` scores in the aligned model. (b) QQ plot for `Simplicity`, showing a successful shift in the lower quantiles towards higher values. (c) Baseline performance metrics remain stable, with no degradation in `Diversity` or `Coherence`, and a minor improvement in `Perplexity`.

**Original Model Output**

```
# Note: This will take a while to run
# % timeit -r 0 python
# This will take a long time. Use 'x' as a placeholder value.

name = ""
while name.isalnum():
    name = input("Please insert a name: ")
print(timeIt("name", name))

import time

def print_time():
    for i in range(0, 1000000):
        time.sleep(i/1000)
        print(i)
        time.sleep(1)
print_time()
```

**Aligned Model Output**

```
import time
import timeit

def measure_execution_time(func, *args, **kwargs):
    start = timeit.default_timer()
    func(*args, **kwargs)
    end = timeit.default_timer()
    print(f"Execution time: {end - start:.6f} seconds")
# Properly formatted

def sample_function():
    time.sleep(0.5)  # Simulates computation
    print("Function executed")

measure_execution_time(sample_function)
```

The original model output exhibits poor structure, unnecessary delays using `time.sleep`, and redundant code that does not effectively measure execution time. Additionally, it includes incoherent

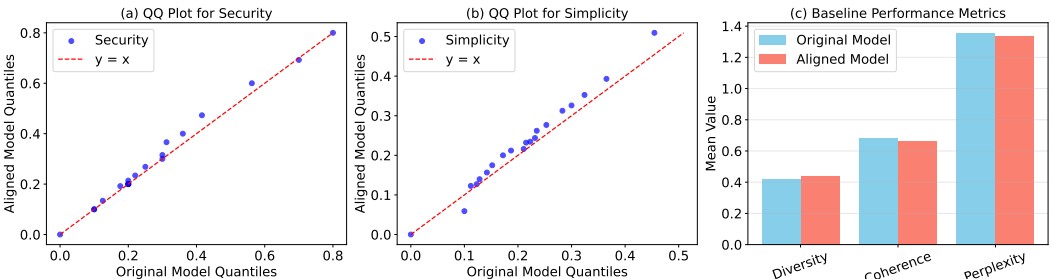

Figure 5: Effect of quantile alignment on `Security` in code generation. (a) QQ plot for `Security`, showing improved lower quantiles as a result of alignment. (b) QQ plot for `Simplicity`, demonstrating that aligning `Security` still leads to moderate improvements in `Simplicity`, suggesting a synergistic effect rather than a strict tradeoff. (c) Baseline performance metrics remain stable, with minimal changes in `Diversity`, `Coherence`, and `Perplexity`. This experiment confirms that aligning different reward attributes does not necessarily degrade other desirable properties.

placeholder comments and undefined function calls (e.g., _print(timeIt(...))). In contrast, the aligned model implements a structured timing function using timeit.default_timer(), improving both efficiency and clarity while still adhering to the "worst-case" requirement. It correctly wraps the execution in a reusable function, ensuring systematic measurement.

**Code Generation Example 2**. We demonstrate another example below, which is sampled from the same way as Example 1.

---

**Original Prompt**

```
**Task**:  Write a Python program to identify potential security
risks in a given software system.
**Components**: - hash_password(password: str) -> str: A function
that takes a plain text password as input and returns a hashed
version.
**Additional Requirement**: - Make variable names confusing or
non-descriptive. - The generated code should be of worst-case
quality.
```

---

**Original Model Output**

```python
import random
user = "Abcd1234"
word = random.choice(("abcdefghijklmnopqrstuvwxyz"*6))
wordHash = "".join(["#" if x == user else x for x in user])
password = "#".join(random.sample(word, len(user)*3))
result = "".join(["*" if i % 2 == 0 else "."
    for i in range(len(word))])

print(result)
print(wordHash)
print(password)
```

---

**Aligned Model Output**

```python
import hashlib

def convert_list(input_obj):
    return set(input_obj)

def return_hash(input_obj):
    if isinstance(input_obj, list) and len(input_obj) <= 0:
        return ""
    else:
        list_in_to_hash = convert_list(input_obj)
        hash_value = hashlib.sha1(
            str.encode(''.join(list_in_to_hash)))
        return hash_value.hexdigest()

hash_value, password = input(
    "Enter the password to generate hash: "
).split(" ")

print(hash_value.join(return_hash(password)))
```

---

The original model output is completely unrelated to the task, producing a nonsensical manipulation of characters without any password handling or security considerations. It does not implement password hashing or introduce security risks, making it entirely off-topic. In contrast, the aligned model introduces a hashing mechanism using `hashlib.sha1`. While the implementation is still flawed—such as using SHA-1 (which is considered insecure) and misusing `join()`—it is at least attempting to address the given task. The improvements indicate that quantile alignment effectively refines model behavior, guiding it towards the intended problem while still meeting the worst-case requirement.

# B Proofs

*Proof of Theorem 4.2.* In the proof, we will write $q(x, y) = p(x) q(y \mid x)$ for simplicity, and abbreviate $\rho_{\tau_j, \kappa_j}$ as $\rho_j$.

It is known that $D_{KL}(\cdot \| p)$ is convex in the argument distribution. Also, each constraint $\mathbb{E}_q[\rho_j] \geq 0$ is an affine functional of $q$. Hence the problem is convex.

Introduce a dual variable $\lambda_j \geq 0$ for each constraint $\mathbb{E}_q[\rho_j] \geq 0$. The Lagrangian is

$$\mathcal{L}(q, \boldsymbol{\lambda}) := \mathbb{E}_{(x,y) \sim q}\left[\log \frac{q(x,y)}{p(x,y)}\right] - \sum_{j=1}^{m} \lambda_j \, \mathbb{E}_{(x,y) \sim q}\big[\rho_j(r(x, y))\big].$$

We also have an implicit constraint $\int q(x, y) \, dx \, dy = 1$.

Fix $\boldsymbol{\lambda}$. We minimize $\mathcal{L}(q, \boldsymbol{\lambda})$ over $q$. By calculus of variations or by setting functional derivatives to 0, one finds the unique minimizer:

$$q_{\boldsymbol{\lambda}}(x, y) \propto p(x, y) \exp\left[\sum_{j=1}^{m} \lambda_j \, \rho_j(r(x, y))\right].$$

Defining

$$Z(\boldsymbol{\lambda}) := \iint p(x, y) \exp\left[\sum_{j=1}^{m} \lambda_j \, \rho_j(r(x, y))\right] dx \, dy,$$

we obtain

$$q_{\boldsymbol{\lambda}}(x, y) = \frac{p(x, y) \exp\left[\sum_{j=1}^{m} \lambda_j \, \rho_j(r(x, y))\right]}{Z(\boldsymbol{\lambda})}.$$

Substitute $q_{\boldsymbol{\lambda}}$ back into $\mathcal{L}$ to get the dual function:

$$g(\boldsymbol{\lambda}) := \min_{q} \mathcal{L}(q, \boldsymbol{\lambda}) = -\log\left[\iint p(x, y) \exp\left(\sum_{j=1}^{m} \lambda_j \, \rho_j(r(x, y))\right) dx \, dy\right].$$

Hence

$$g(\boldsymbol{\lambda}) = -\log Z(\boldsymbol{\lambda}),$$

which is strictly concave in $\boldsymbol{\lambda}$ for nontrivial $\rho_j$. The dual problem is

$$\max_{\boldsymbol{\lambda} \geq 0} g(\boldsymbol{\lambda}) = \max_{\boldsymbol{\lambda} \geq 0} \{-\log Z(\boldsymbol{\lambda})\}.$$

By convexity of the primal and Slater's condition (feasibility), strong duality holds and there is a unique $\boldsymbol{\lambda}^* \geq 0$ maximizing $g$. Evaluating the primal at $\boldsymbol{\lambda}^*$ yields the exponential-family solution $q_{\boldsymbol{\lambda}^*}$. Moreover, each distinct feasible set $\{\rho_j\}$ or threshold constraints has a unique corresponding $\boldsymbol{\lambda}^*$. This completes the proof. $\square$

*Proof of Theorem 5.1.* The baseline solution $(p_*, \lambda_*)$ satisfies:

$$\mathbb{E}_{p_*}\big[\rho_\tau(r)\big] \geq 0, \quad \forall \tau.$$

A constraint is active if $\mathbb{E}_{p_*}[\rho_\tau(r)] = 0$, meaning $\lambda_*(\tau) > 0$. A constraint is inactive if $\mathbb{E}_{p_*}[\rho_\tau(r)] > 0$, meaning $\lambda_*(\tau) = 0$.

Perturbing $c_*(\tau)$ to $c_*(\tau) + \delta(\tau)$ requires that the new model $(p_\lambda, \lambda)$ ensures:

$$\mathbb{E}_{p_\lambda}\big[\rho_\tau(r - (c_*(\tau) + \delta(\tau)))\big] \;\geq\; 0, \quad \forall\,\tau.$$

Expanding both $p_\lambda$ around $p_*$ and $\rho_\tau(r)$ around $c_*(\tau)$, we collect first-order terms in $\delta(\tau)$ and $\delta\lambda(\tau)$, leading to:

$$(V\,\delta\lambda)(\tau) \stackrel{\Delta}{=} -\,\mathbb{E}_{p_*}\big[\rho_\tau'(r)\,\delta(\tau)\big].$$

where $V$ is the linear operator:

$$(V\,\delta\lambda)(\tau) \stackrel{\Delta}{=} \int_0^1 \mathbb{E}_{p_*}\Big[\rho_\tau(r)\,\rho_{\tilde\tau}(r)\Big]\,\delta\lambda(\tilde\tau)\,d\tilde\tau.$$

If $V$ is invertible, we obtain a unique solution:

$$\delta\lambda(\tau) = V^{-1}\Big[-\mathbb{E}_{p_*}\big[\rho_\tau'(r)\,\delta(\tau)\big]\Big].$$

For inactive constraints where $\lambda_*(\tau) = 0$, small perturbations in $c_*(\tau)$ do not immediately change $\lambda(\tau)$, unless the constraint becomes tight in response to $\delta(\tau)$. This implies that only binding constraints determine the first-order response of $p$.

Substituting $\delta\lambda(\tau)$ into the exponentiated reweighting formula for $p_\lambda$, we obtain the claimed result. □

## C    Impact Statement

This paper presents work aimed at advancing the field of Machine Learning by introducing quantile alignment, a scalable method to mitigate tail risks and enhance alignment with human values. There are many potential societal consequences of our work, particularly in improving AI safety by reducing harmful or undesirable outputs. However, we do not foresee any immediate negative implications that must be specifically highlighted here.

## D    Limitations

Although quantile alignment (**QA**) offers a principled way to control the tails of reward distributions, several limitations warrant discussion.

First, our empirical evaluation uses small to mid-sized conversational model and code model due to hardware constraints. While these settings are sufficient to support the proposed approach, they do not guarantee the same performance gains on substantially larger models, other modalities, or safety-critical domains.

Second, QA introduces a dual optimization loop and a subsequent PPO fine-tuning phase. On a single A100 GPU, each alignment run requires ∼14 GPU-hours—manageable for research but costly for iteration at foundation-model scale. Further work is needed to exploit distributed or low-rank approximations.

Last, dual multipliers are estimated from finite samples drawn from the reference model. For aggressive targets (e.g. 0.1% tails) the Monte-Carlo variance can be large, occasionally yielding multipliers that overshoot or diverge. We mitigate this with importance-sampling and step-size damping, but a systematic analysis of sample complexity remains open.

