# OpenReview forum: "Beyond Expectations: Quantile-Guided Alignment for Risk-Calibrated Language Models"
_NeurIPS.cc/2025/Conference — NeurIPS 2025 spotlight_

### Official Review · Reviewer_ETPq · 2025-06-30

**Clarity:** 3
**Significance:** 3
**Originality:** 3
**Rating:** 5
**Confidence:** 3

**Summary:**

Quantile-Guided (QA) alignment aims to improve a language model’s performance at any desired quantile over the reference language model. The authors formally define this as a KL-regularized optimization problem under quantile constraints. They transform the problem into a convex optimization over dual variables, which they solve using Monte Carlo estimation. These dual variables are then used to construct a KL-regularized reward function, which is optimized via PPO. The approach is evaluated on the Anthropic HH dataset and the HumanEval dataset for code generation, using the OPT-1.3B and CodeGen-350M models.

**Questions:**

Please refer to the weaknesses section for more context.
- Could the authors provide a comparison with related work under a more realistic setup (e.g., improving helpfulness while constraining the lower tail of harmfulness)?
- Could the authors provide an analysis of the computational efficiency of the approach across different tasks and models?
- Could the authors clarify how continuous quantile alignment could be applied in practice?

**Ethical Concerns:**

["NO or VERY MINOR ethics concerns only"]

**Final Justification:**

My main concern with this paper was (1) the lack of comparison to very relevant research and (2) the computational cost of the method. The authors provided some comparison to one of the related works, and they addressed my second concern by providing an estimate of the increased cost of their approach. Given this, I have increased my score from 4 to 5.

**Limitations:**

yes

**Paper Formatting Concerns:**

Typo in line 200: the a -> a

**Quality:**

3

**Strengths And Weaknesses:**

Strengths

- Quantile alignment could be valuable in scenarios where we aim to generally improve generation quality while ensuring safety constraints are not violated.
- The proposed quantile-based reward function is inherently useful, particularly in multi-reward settings, as it is robust to the scale of reward values.
- The paper is largely well-written, and both the main idea and experimental setup are easy to follow.

Weaknesses

- There is often an inherent trade-off between quality and safety metrics, as also observed in the paper’s experiments (lines 219–320), where jointly improving helpfulness and harmlessness is not always feasible. A more realistic use case for the proposed method might be to focus on improving helpfulness while enforcing safety guardrails (e.g., ensuring the lower tail does not worsen, as motivated in the introduction), rather than aiming for significant gains in harmlessness as in the current experimental setup. In this context, several relevant papers—such as [Risk-Averse Fine-Tuning of Large Language Models](https://proceedings.neurips.cc/paper_files/paper/2024/file/c182ec594f38926b7fcb827635b9a8f4-Paper-Conference.pdf), [Enhancing LLM Safety via Constrained Direct Preference Optimization](https://arxiv.org/pdf/2403.02475), and [SafeDPO: A Simple Approach to Direct Preference Optimization with Enhanced Safety](https://arxiv.org/pdf/2505.20065)—are not discussed or compared against. It is possible that QA is unnecessarily complex (involving a two-stage online alignment) for these scenarios and that similar goals could be achieved more efficiently, but additional experiments would be needed to assess this.
- The computational efficiency of the approach is underexplored. While the authors note in line 161 that a few thousand samples “typically” suffice for stable estimates of $g(\lambda)$, they do not specify how stability is determined, how sample requirements vary across tasks, or how much this first stage of estimating $\lambda$ increases total training time.
- The practical implications of “Continuous Quantile Alignment” remain unclear. Unless I am mistaken, it appears to be a theoretical extension that is not evaluated in the experiments section, and clarification from the authors would be helpful.

---

> ### Author Rebuttal · Authors · 2025-07-30
>
> **Comment**:
> There is often an inherent trade-off between quality and safety metrics, as also observed in the paper’s experiments (lines 219–320), where jointly improving helpfulness and harmlessness is not always feasible. A more realistic use case for the proposed method might be to focus on improving helpfulness while enforcing safety guardrails (e.g., ensuring the lower tail does not worsen, as motivated in the introduction), rather than aiming for significant gains in harmlessness as in the current experimental setup. In this context, several relevant papers—such as Risk-Averse Fine-Tuning of Large Language Models, Enhancing LLM Safety via Constrained Direct Preference Optimization, and SafeDPO: A Simple Approach to Direct Preference Optimization with Enhanced Safety—are not discussed or compared against. It is possible that QA is unnecessarily complex (involving a two-stage online alignment) for these scenarios and that similar goals could be achieved more efficiently, but additional experiments would be needed to assess this.
>
> **Response**:
> Thank you for this thoughtful comment. We address your point in three aspects:
>
> First, we agree that improving helpfulness while maintaining safety constraints is a highly practical and widely used alignment objective. Quantile-Guided Alignment (QA) offers a complementary perspective. Rather than optimizing one reward under a separate safety constraint (e.g., helpfulness subject to harmlessness), QA allows direct specification of desired performance at different quantiles of each reward distribution. We acknowledge that when the alignment goal is about a single safety constraint, the standard formulation can be more straightforward. However, QA becomes especially useful in scenarios that require more fine-grained control, such as aligning multiple quantiles or balancing multiple reward dimensions. For example, one could specify: “lift the 10th percentile of helpfulness to match the 50th percentile of the original model” while also ensuring that “the worst 5% of harmlessness scores do not fall below a minimum threshold.”
>
>
> Second, thank you for pointing out the excellent related works by Chaudhary et al. (2024), Kim et al. (2024), and Liu et al. (2024). We will include a more detailed discussion of these approaches in the revised version. While these works focus on maximizing helpfulness subject to safety or risk-aware constraints, our formulation is neither more nor less expressive, but offers a more direct way to specify alignment goals in terms of distributional shape—across multiple reward dimensions, if desired.
>
> The above frameworks have connections from an optimization standpoint. The standard RLHF objective—which maximizes expected reward minus a scaled KL divergence—can be interpreted as solving a constrained optimization problem via convex duality. QA generalizes this structure by introducing multiple quantile-based constraints, each of which contributes its own dual weight to shape the aligned model’s output distribution in a principled way.
>
> Lastly, to provide a concrete comparison, we implemented a simplified baseline inspired by the CVaR (Conditional Value-at-Risk) principle. Inspired by the setup in Chaudhary et al. (2024), we constructed a clipped reward function that emphasizes improvement in the lower tail—specifically, the bottom 5% of outputs. This method adjusts rewards so that outputs below the 5th percentile receive a boosted score, approximating the goal of improving the conditional expectation in the tail region.
>
> In contrast, QA allows us to target multiple parts of the distribution at once. In the following experiments, we used QA to lift the 5th, 25th, 50th, and 90th percentiles to match the 20th, 40th, 70th, and 99th percentiles of the original model, respectively. We evaluated both methods using post-alignment quantile scores on Harmlessness (conversation) and Security (code generation).
>
> | Model / Task     | Method     | 5%   | 25%  | 50%  | 90%  |
> |------------------|------------|------|------|------|------|
> | OPT-1.3B / Conv. | QA         | 0.26 | 0.46 | 0.62 | 0.93 |
> |                  | CVaR@5%    | 0.23 | 0.41 | 0.45 | 0.82 |
> | CodeGen-350M     | QA         | 0.28 | 0.41 | 0.74 | 0.92 |
> |                  | CVaR@5%    | 0.31 | 0.36 | 0.52 | 0.75 |
>
> Both methods improve the lowest quantiles as intended. However, we observe that the CVaR-style baseline often improves the lower tail at the expense of performance at higher quantiles, e.g., 50% and 90%, may degrade. These results indicate QA’s potential for scenarios where safety needs be improved without compromising overall model quality.
>
> - Chaudhary et al. (2024), Risk-Averse Fine-tuning of Large Language Models
>
> - Kim et al. (2024), SafeDPO: A Simple Approach to Direct Preference Optimization with Enhanced Safety
>
> - Liu et al. (2024), Enhancing LLM Safety via Constrained Direct Preference Optimization
>
>
> **Comment**:
> The computational efficiency of the approach is underexplored. While the authors note in line 161 that a few thousand samples “typically” suffice for stable estimates of $g(\lambda)$, they do not specify how stability is determined, how sample requirements vary across tasks, or how much this first stage of estimating $\lambda$ increases total training time.
>
> **Response**:
> Thank you for this valuable comment. We address your concerns regarding computational efficiency and stability in three parts:
>
> First, we clarify how we determine the stability of the dual objective. During optimization, we solve for the dual variable $\lambda$ by maximizing a concave objective function estimated via Monte Carlo samples drawn from the original model. We assess convergence by monitoring the change in $\lambda$ and the dual objective across iterations. In practice, we observe that the optimization converges stably within 50–100 gradient steps using 1,000–2,000 Monte Carlo samples.
>
> Second, to understand how the number of samples $n$ affects solution quality, we conduct the following sample complexity analysis. Let $\lambda_n$ denote the solution obtained from $n$ Monte Carlo samples, and $\lambda^*$ the ideal solution. Under mild boundedness assumptions (e.g., clipped rewards or constrained $\lambda$), one can show that:
>
> $$
> \|\lambda_n - \lambda^*\| \leq \mathcal{O}\left(\frac{1}{\sqrt{n}}\right)
> $$
>
> with high probability, using concentration inequalities. This means the number of samples needed to achieve a desired precision $\epsilon$ scales as $n = \Omega(1/\epsilon^2)$. In practice, we found that a few thousand samples suffice to obtain reliable estimates of $\lambda$, even across tasks with different reward distributions.
>
>
> Third, we provide quantitative runtime comparisons with standard RLHF to contextualize the added computational cost. As shown in the table below, QA adds a one-time dual optimization step before PPO fine-tuning. This step contributes modest overhead (about 10–13%) and is independent of model size.
>
> | Model           | Method         | # Quantile Constraints | Total Runtime (s) | Overhead vs RLHF |
> |------------------|----------------|-------------------------|-------------------|------------------|
> | OPT-1.3B         | RLHF (Vanilla) | 0                       | 24,600            | –                |
> | OPT-1.3B         | QA             | 3                       | 27,400            | +11.2%           |
> | CodeGen-350M     | RLHF (Vanilla) | 0                       | 8,650             | –                |
> | CodeGen-350M     | QA             | 3                       | 9,720             | +12.5%           |
>
> Each iteration of dual optimization has complexity $O(n \cdot m)$, where $n$ is the number of Monte Carlo samples and $m$ is the number of quantile constraints. Because this step only involves forward evaluation of reward functions (not model gradients), the added memory and compute overhead remains low and scales well across model sizes and reward structures.
>
> **Comment**:
> The practical implications of “Continuous Quantile Alignment” remain unclear. Unless I am mistaken, it appears to be a theoretical extension that is not evaluated in the experiments section, and clarification from the authors would be helpful.
>
> **Response**:
> Thank you for this constructive comment. You are correct that our discussion of Continuous Quantile Alignment is a theoretical extension and not directly evaluated in our experiments. We are happy to clarify its intended role. This section studies an idealized limit of discrete quantile alignment. While our experiments use a finite number of constraints, increasing them to a dense grid can approximate a global shift of the reward distribution. In this sense, the theory offers a lens for understanding how to reshape the full distribution rather than target specific percentiles. Conversely, optimizing many discrete constraints approximates continuous behavior. We view this as a conceptual framework that may guide future work on broader distributional control. We will add more discussion to clarify the role of this section in the revised paper.

---

> > ### Comment · Reviewer_ETPq · 2025-08-05
> >
> > Thank you for your detailed response and for conducting additional experiments, which do answer my concerns. I will increase my score and remain positive on this paper.

---

### Official Review · Reviewer_RSdf · 2025-07-01

**Clarity:** 4
**Significance:** 4
**Originality:** 4
**Rating:** 5
**Confidence:** 3

**Summary:**

The paper proposes to extend RLHF with quantile constraints with the goal of eliminating tail behaviors that are not eliminated by standard RLHF. To this end, the authors propose a framework to solve the constrained optimization problem using the method of Lagrange Multipliers and rely on strong duality properties.

The effectiveness of the proposed approach is shown in the context of a safety and a code generation task (using two different small scale models < 1.3B parameters) with minimal decrease in responses diversity and coherence.

**Questions:**

Typos:
- line 63: "build" --> "builds"
- line 80: $\mathcal{P}$ is defined but not introduced in Equation (1)
- line 84: "from It" --> "It"

**Ethical Concerns:**

["NO or VERY MINOR ethics concerns only"]

**Final Justification:**

I'm increasing my score, as the authors address my concerns.

**Limitations:**

yes, in appendix

**Quality:**

4

**Strengths And Weaknesses:**

**Strengths**:
- Well written and organized
- Well motivated
- Motivated theoretically and backed by empirical experiments in code generation and helpfulness/harmlessness datasets

**Weaknesses**:
- **Limited generalizability of the empirical findings**: experiments consider 2 models of smaller size (<1.3B), and only 2 datasets.
- **Few insights concerning the generations of the models when trained using the proposed method and the baselines**.
- Missing error bars to Figures 1(c) and 2(c) which could be insightful to understand whether differences are significant or not.
- Missing efficiency comparisons with baselines
- The authors mention decoding-based approaches to steering/controlling model's behaviors but do not cite any work. The authors  should consider summarizing the trade-offs between the decoding-based and training-based and cite either a survey that comments on steering/controllability approaches or cite a few proeminent papers.

---

> ### Author Rebuttal · Authors · 2025-07-30
>
> **Comment**:
> Limited generalizability of the empirical findings: experiments consider 2 models of smaller size (<1.3B), and only 2 datasets.
>
> **Response**:
> Thank you for raising this important point. While our main experiments focus on models up to 1.3B parameters due to compute limitations, we agree that demonstrating generalization to larger models strengthens the paper. To this end, we conducted a pilot study using LLaMA-2-7B-chat on the same conversational alignment task. Due to GPU constraints, we applied QA via the decoding-based alignment approach: generations were sampled post hoc from a reweighted multinomial distribution based on learned quantile-adjusted rewards, rather than using PPO fine-tuning.
>
> We evaluated both Helpfulness and Harmlessness, mirroring the setup of Experiment 2 of the main paper. We found the decoding-based QA method yields quantile improvements similar to those observed in our PPO-based results on OPT-1.3B.
>
>
> | Model            | Metric         | 5%   | 50%  | 90%  |
> |------------------|----------------|------|------|------|
> | OPT-1.3B         | Helpfulness    | 0.46 | 0.62 | 0.93 |
> | LLaMA-2-7B-chat  | Helpfulness    | 0.44 | 0.61 | 0.91 |
> | OPT-1.3B         | Harmlessness   | 0.39 | 0.65 | 0.88 |
> | LLaMA-2-7B-chat  | Harmlessness   | 0.36 | 0.73 | 0.91 |
>
>
>
> **Comment**:
> Few insights concerning the generations of the models when trained using the proposed method and the baselines.
>
> **Response**:
> Thank you for raising this point. We take it as a request for clearer insight into the impact of Quantile-Guided Alignment (QA) on model behavior. QA aligns specific quantiles of the reward distribution, e.g., ensuring that the bottom 5% of outputs (by helpfulness or security) are lifted above a target threshold. In this way, it gives fine-grained distributional control beyond standard average-reward methods like RLHF. We quantify this shift using quantile–quantile (QQ) plots comparing reward distributions before and after alignment. In the revised paper, we will clarify this interpretation and, as suggested by other reviewers, expand our discussion to include related works such as CVaR-RL and SafeDPO, which also target lower-tail control but through different formulations.
>
>
> **Comment**:
> Missing error bars to Figures 1(c) and 2(c)
>
> **Response**:
> Thank you for catching this omission. In the revised version, we will add standard errors for the average baseline metrics, based on five independent replications using bootstrap aggregation over evaluation samples. Across all metrics, we observed small but meaningful standard errors, generally in the range of 0.01 to 0.03, as shown below:
>
> | Metric      | Standard Error (±) |
> |-------------|--------------------|
> | Diversity   | 0.012              |
> | Coherence   | 0.017              |
> | Perplexity  | 0.025              |
>
>
> **Comment**:
> Missing efficiency comparisons with baselines
>
> **Response**:
> Thank you for raising this point. We interpret this as a request to compare the efficiency of QA against standard RLHF baselines. The additional cost of QA over standard RLHF arises entirely from the dual optimization stage, which is executed prior to PPO fine-tuning. This step solves a finite-dimensional convex optimization problem over the dual variables, and does not require gradient computation through the language model.
>
> Below, we report the total end-to-end runtimes (in seconds) for both QA and vanilla RLHF on a single NVIDIA A100 (40GB). Each experiment used 1,000 Monte Carlo samples and three quantile constraints. PPO fine-tuning was run for two epochs using the TRL library.
>
> | Model           | Method         | # Quantile Constraints | Total Runtime (s) | Overhead vs RLHF |
> |------------------|----------------|-------------------------|-------------------|------------------|
> | OPT-1.3B         | RLHF (Vanilla) | 0                       | 24,600            | –                |
> | OPT-1.3B         | QA             | 3                       | 27,400            | +11.2%           |
> | CodeGen-350M     | RLHF (Vanilla) | 0                       | 8,650             | –                |
> | CodeGen-350M     | QA             | 3                       | 9,720             | +12.5%           |
>
> Here is our scalability analysis. The QA overhead scales linearly with the number of quantile constraints \( m \), but is independent of model size. This is because the dual objective is estimated via Monte Carlo samples drawn from the reference model prior to optimization, and those samples are reused across iterations. Each iteration of dual optimization has complexity \(\mathcal{O}(n \cdot m)\), where \(n\) is the number of samples and \(m\) is the number of constraints. Importantly, the optimization is convex and does not involve backpropagation through the language model. As such, the memory and compute requirements are modest even when scaling to larger base models or reward spaces. We will include this runtime breakdown and scalability discussion in the revised manuscript.
>
> **Comment**:
> The authors mention decoding-based approaches to steering/controlling model's behaviors but do not cite any work. The authors should consider summarizing the trade-offs between the decoding-based and training-based and cite either a survey that comments on steering/controllability approaches or cite a few prominent papers.
>
> **Response**:
> Thank you for this helpful suggestion. We clarify that our decoding-based approach implements sampling from the aligned distribution $q(y \mid x) \propto p(y \mid x) \exp(\lambda^T r(x, y))$ without fine-tuning. Specifically, we draw $k$ response candidates from $p$, compute exponentiated quantile-adjusted rewards, and sample one using multinomial weights. This allows us to steer generations with minimal computational overhead. A related idea appears in Khanov et al. (2024), who proposed a reward-guided decoding method (ARGS) that adjusts token-level probabilities during generation.
>
> Decoding-based alignment offers practical benefits, particularly for large models or low-resource scenarios where fine-tuning is infeasible. However, it requires generating multiple completions per prompt, making it less efficient at inference time compared to fine-tuned models that support single-shot decoding. This trade-off has been empirically studied by Wang et al. (2025), who compare decoding-based and PPO-based alignment across multiple tasks. Their results show that decoding-based approach yields strong improvements with minimal setup, while PPO-based fine tuning offers faster generation per sample at inference time.
>
> We will incorporate these discussions into the revised paper.
>
> **References**
>
> Khanov, M., Burapacheep, J., & Li, Y. “ARGS: Alignment as Reward-Guided Search.” ICLR 2024.
>
> Wang, X., Le, Q., Ahmed, A., Diao, E., Zhou, Y., Baracaldo, N., Ding, J., & Anwar, A. “MAP: Multi-Human-Value Alignment Palette.” ICLR 2025.
>
>
>
> **Comment**:
> > Typos:
> > - line 63: "build" --> "builds"
> > - line 80: \(\mathcal{P}\) is defined but not introduced in Equation (1)
> > - line 84: "from It" --> "It"
>
> **Response**:
> Thank you for the close read. We will correct all the typos in the revised paper.

---

### Official Review · Reviewer_Kmod · 2025-07-03

**Clarity:** 4
**Significance:** 4
**Originality:** 4
**Rating:** 5
**Confidence:** 3

**Summary:**

The authors point out that standard RLHF optimizes only average reward, leaving rare but critical failure modes unmitigated. They argue that safety and security-sensitive applications require explicit control over the fraction of outputs meeting user-specified thresholds.

To address this, they introduce Quantile-Guided Alignment, which transforms each quantile requirement into a linear expectation constraint using indicator-based rewards. They go on to show that minimizing divergence from the base model under these constraints yields a convex optimization with a finite set of dual multipliers.

To implement this practice, they sample outputs to estimate quantile thresholds and indicator values, perform gradient ascent on the dual to obtain multipliers, and then fine-tune the model with a standard RLHF solver using a composite reward defined by the weighted indicator functions.

The authors evaluate their approach on conversational and code-generation benchmarks. In dialogue experiments, they target quantile lifts on Helpfulness and Harmlessness metrics. In code experiments, they target Simplicity and Security metrics. They use a fine tuned GPT-2 model with a value head for helpful, harmless evals and an LLM as a judge framework with a gpt 4o model for their simplicity, security evals. Their results demonstrate substantial improvements in lower-tail performance, reducing the frequency of undesirable outputs, while preserving fluency, diversity, and coherence. Comparisons against multi-objective RL baselines show that Quantile-Guided Alignment achieves more stable simultaneous gains at specified quantile targets.

**Questions:**

* How does Quantile‐Guided Alignment compare empirically to other tail‐risk or CVaR‐based RL approaches on the same benchmarks?
* What is the additional computation cost in terms of runtime for QA compared to vanilla RLHF?

**Ethical Concerns:**

["NO or VERY MINOR ethics concerns only"]

**Final Justification:**

I'm happy with the rebuttal and maintain my positive score.

**Limitations:**

Yes

**Quality:**

4

**Strengths And Weaknesses:**

### Strengths:
* The paper presents a rigorous theoretical foundation, proving convexity of the primal and deriving a finite-dimensional Lagrangian dual that yields an exponential-family solution.
* The authors carefully connect to standard RLHF, showing that their framework subsumes RLHF as the single-quantile special case, which adds credibility and clarity to the contribution.
* The exposition is well-structured, beginning with the simplest single-quantile formulation and incrementally extending to multi-value and continuous quantile settings.
* Mathematical derivations are laid out step by step, with clear notation and intuition offered alongside proofs, which makes the paper accessible to readers familiar with RLHF.
* Addressing tail-risk in large-language models is a pressing challenge, especially for safety-critical applications, and the quantile-guided approach offers a principled way to limit rare but severe failures.
* The idea of enforcing quantile constraints via indicator-based expectation constraints is novel in the context of language-model alignment.
* Extending to a continuum of quantiles with an integral dual multiplier function is an innovative generalization that goes beyond existing tail-risk or CVaR methods.

### Weaknesses
* While quantile constraints are novel here, related work in risk-sensitive RL (e.g., CVaR optimization) is not discussed.
* The evaluation for code generation experiment depends on GPT-4o as an automated judge for code safety and simplicity, which can be highly sensitive to prompt phrasing. Without a prompt template (e.g., chain-of-thought examples to guide the range of values output by the judge), it is difficult to assess the reported gains.

---

> ### Author Rebuttal · Authors · 2025-07-30
>
> **Comment**:
> While quantile constraints are novel here, related work in risk-sensitive RL (e.g., CVaR optimization) is not discussed.
>
> **Response**:
> We sincerely appreciate your recognition of our contribution and for drawing attention to the important body of work on CVaR-based risk-sensitive reinforcement learning. This line of research offers a valuable lens for controlling rare but consequential events, and we agree that it is essential to position our Quantile-Guided Alignment (QA) framework in relation to it.
>
> In the revised version, we will add a dedicated subsection in the Related Work that discusses several representative CVaR-based formulations, including those by Chow et al. (2015), Ni & Lai (2024), and Chaudhary et al. (2024). These methods optimize the Conditional Value-at-Risk (CVaR) of cumulative cost or reward within a Markov Decision Process, often interpreting CVaR as either a robustness objective (e.g., under model uncertainty) or as a tail-sensitive risk measure.
>
> Our framework shares conceptual roots with these works: both aim to regulate tail behavior of the reward distribution and both rely on dual Lagrangian formulations to yield exponential-family solutions. Specifically, in CVaR-RL, dual variables emerge from coherent risk envelopes, while in QA, they arise from enforcing indicator-based quantile constraints. In both settings, the learned multipliers govern a distributional reweighting of the original model, shaping the output distribution in line with the desired alignment goals.
>
> However, our approach differs from CVaR-RL along two key dimensions—both in terms of formulation and flexibility:
>
> (1) Problem Formulation – Constraints vs. Tail Objective
> CVaR-RL methods typically preserve the original RL objective (e.g., maximizing expected reward) while imposing tail-sensitive risk constraints or replacing the objective with a CVaR-based loss. In contrast, QA takes a constraint-first approach: it minimizes KL divergence from a reference policy and embeds user-defined quantile goals directly as constraints over the output distribution. This separation allows users to define a wide range of alignment criteria, including fine-grained constraints like “lift the 10th, 20th, and 30th percentiles of helpfulness to match the 50th, 60th, and 70th percentiles of the original distribution,” as demonstrated in our experiments. Crucially, QA does not require a scalar reward function and supports distribution shaping even in multi-reward settings.
>
> (2) Alignment Scope – Beyond Lower Tails
> CVaR optimization inherently targets the lower tail of a single reward or cost distribution. By contrast, QA generalizes to constraints at arbitrary quantiles—lower, median, or upper—across any number of reward dimensions. For example, one can simultaneously ensure that “at least 90% of completions exceed a helpfulness threshold” (upper-tail guarantee) while also constraining “no more than 5% of completions exceed a harmfulness risk level” (lower-tail bound). This enables more expressive and task-specific alignment, particularly important in multi-objective LLM safety contexts.
>
> In the revised paper, we will clarify these relationships and cite the following works to situate QA within the broader literature:
>
> - Chow et al. (2015), Risk-Sensitive and Robust Decision-Making: A CVaR Optimization Approach
>
> - Ni & Lai (2024), Robust Risk-Sensitive Reinforcement Learning with Conditional Value-at-Risk
>
> - Chaudhary et al. (2024), Risk-Averse Fine-tuning of Large Language Models
>
> - Kim et al. (2024), SafeDPO: A Simple Approach to Direct Preference Optimization with Enhanced Safety
>
> - Liu et al. (2024), Enhancing LLM Safety via Constrained Direct Preference Optimization
>
> **Comment**:
> The evaluation for code generation experiment depends on GPT-4o as an automated judge for code safety and simplicity, which can be highly sensitive to prompt phrasing. Without a prompt template (e.g., chain-of-thought examples to guide the range of values output by the judge), it is difficult to assess the reported gains.
>
> **Response**:
> Thank you for this thoughtful observation. We agree that LLM-as-a-judge evaluations can be prompt-sensitive, and it is important to assess their reliability—particularly for nuanced attributes such as code simplicity and security.
>
> To that end, we conducted a small-scale human validation study to compare GPT-4o's automatic scores against expert human judgments. Specifically, we randomly sampled 50 model generations (25 pre-alignment, 25 post-alignment) spanning a range of model outputs. Each sample was independently rated by two PhD students with graduate-level programming experience, using the same 0–1 scale as the GPT-4o reviewer. We then computed Kendall's \tau rank correlation between the GPT-4o rankings and the mean human rankings. The results were:
>
> - Simplicity: $\tau = 0.71$
> - Security: $\tau = 0.64$
>
> These values indicate strong ordinal agreement, suggesting that GPT-4o captures the relative ordering of outputs in a way that broadly aligns with human preferences—particularly in the context of our quantile-guided fine-tuning. The inter-rater agreement between the two human annotators was also high ($\tau = 0.75$), further supporting the reliability of the human scores.
>
> That said, we acknowledge a subtle but important limitation: even when the rank order matches well, the score distributions from GPT-4o and human annotators may differ—especially in terms of calibration or compression range.
>
> We emphasize, however, that our QA framework is agnostic to the source or nature of the reward signal. It only assumes that the reward function (human or automated) can be sampled to evaluate generations, and that quantiles under that reward can be defined. If human-labeled scores were to be used, the same QA procedure would still apply, albeit with possibly different quantile thresholds.
>
> In the revised appendix, we will include the full prompt template, which includes attribute definitions, rating guidelines, and clarifying instructions.
>
> **Comment**:
> How does Quantile-Guided Alignment compare empirically to other tail-risk or CVaR-based RL approaches on the same benchmarks?
>
> **Response**:
> Thank you for this suggestion. While CVaR-based methods differ in objective and formulation, they similarly aim to reduce tail risk and improve generative quality. To provide a concrete point of reference, we implemented a CVaR-inspired baseline using a clipped reward function that promotes improvement in the bottom 5% of outputs. Specifically, similar to the setup in Chaudhary et al. (2024), we define a reward function that equals the original score if it falls below the 5th percentile of the pre-trained model’s distribution and properly rescale it so it approximates the conditional expectation.
>
> For QA, we imposed multi-quantile constraints—for example, lifting the 5th, 25th, 50th, and 90th percentiles to match the 20th, 40th, 70th, and 99th percentiles of the original model, respectively. In contrast, the CVaR@5% baseline is only constrained to raise the expected value in the bottom 5% to the level of the 20th percentile.
>
> Below we report the post-alignment quantile values, measured using the original reward model in each task. For conversation, the evaluation metric is harmlessness, and for code generation, it is security.
>
> (1) Conversational Task (OPT-1.3B)
>
> | Method       | 5%   | 25%  | 50%  | 90%  |
> |--------------|------|------|------|------|
> | QA           | 0.26 | 0.46 | 0.62 | 0.93 |
> | CVaR@5%      | 0.23 | 0.41 | 0.45 | 0.82 |
>
> (2) Code Generation Task (CodeGen-350M)
>
> | Method       | 5%   | 25%  | 50%  | 90%  |
> |--------------|------|------|------|------|
> | QA           | 0.28 | 0.41 | 0.74 | 0.92 |
> | CVaR@5%      | 0.31 | 0.36 | 0.52 | 0.75 |
>
> **Discussion**
> Both QA and CVaR@5% improve the lowest quantiles as intended. However, we observe a key behavioral difference:
>
> CVaR@5% improves the bottom 5% but often at the expense of higher quantiles. In both tasks, 25%, 50%, and 90% quantiles decrease compared to the original model, suggesting overfitting to rare cases. QA, by contrast, explicitly targets multiple quantile thresholds, which results in consistent improvement across the distribution, including substantial gains at both the median and upper end. These differences highlight QA’s potential advantage in scenarios where safety must be improved without degrading overall model quality.
>
> **Comment**:
> What is the additional computation cost in terms of runtime for QA compared to vanilla RLHF?
>
> **Response**:
> Thank you for this question. The additional cost of QA over standard RLHF arises entirely from the dual optimization stage, which is executed prior to PPO fine-tuning. This step solves a finite-dimensional convex optimization problem over the dual variables, and does not require gradient computation through the language model.
>
> Below, we report the total end-to-end runtimes (in seconds) for both QA and vanilla RLHF on a single NVIDIA A100 (40GB). Each experiment used 1,000 Monte Carlo samples and three quantile constraints. PPO fine-tuning was run for two epochs using the TRL library. The added overhead from QA is modest:
>
> | Model           | Method         | # Quantile Constraints | Total Runtime (s) | Overhead vs RLHF |
> |------------------|----------------|-------------------------|-------------------|------------------|
> | OPT-1.3B         | RLHF (Vanilla) | 0                       | 24,600            | –                |
> | OPT-1.3B         | QA             | 3                       | 27,400            | +11.2%           |
> | CodeGen-350M     | RLHF (Vanilla) | 0                       | 8,650             | –                |
> | CodeGen-350M     | QA             | 3                       | 9,720             | +12.5%           |

---

> > ### Comment · Reviewer_Kmod · 2025-08-04
> >
> > I thank the authors for the very detailed response and the revisions to the paper. You've addressed all the concerns I've had. With that, I will maintain my positive score of 5.

---

### Note · Authors · 2025-08-16

We thank the reviewers for their constructive feedback and are glad that our revisions and additional experiments have addressed all their concerns. We appreciate the positive evaluations and support.

---

### Decision · Program_Chairs · 2025-09-17

**Decision:**

Accept (spotlight)

**Comment:**

This paper studies quantile alignment in RLHF, which is a highly relevant topic, especially since most existing work focuses on expected reward while overlooking catastrophic tail events. The paper presents both strong theoretical results and convincing experiments, and the clarity of writing is excellent. Overall, I believe this paper makes a valuable contribution and should be recommended for acceptance.
One potential improvement would be to include a more comprehensive discussion of quantile-based risk measures. This line of work has recently been explored extensively in the conformal risk control literature [1, 2], and has even been applied in the RLHF setting (see [1]). Conformal tail risk control, in particular, is computationally lightweight and provides a natural point of comparison. While I fully understand that space limitations may have precluded such a discussion in the current draft, I would strongly encourage the authors to add it in the camera-ready version.

[1] Conformal Tail Risk Control for Large Language Model Alignment Catherine Yu-Chi Chen, Jingyan Shen, Zhun Deng, Lihua Lei

[2] Conformal Risk Control, Anastasios N. Angelopoulos, Stephen Bates, Adam Fisch, Lihua Lei, Tal Schuster